# Therapy Implications of Hepatitis C Virus Genetic Diversity

**DOI:** 10.3390/v13010041

**Published:** 2020-12-29

**Authors:** Miguel Angel Martinez, Sandra Franco

**Affiliations:** Miguel Angel Martínez, IrsiCaixa, Hospital Universitari Germans Trias i Pujol, 08916 Badalona, Spain; sfranco@irsicaixa.es

**Keywords:** HCV, genetic diversity, quasispecies, therapy

## Abstract

Hepatitis C virus (HCV) is an important human pathogen with a high chronicity rate. An estimated 71 million people worldwide are living with chronic hepatitis C (CHC) infection, which carries the risk of progression to hepatic fibrosis, cirrhosis, and hepatocellular carcinoma (HCC). Similar to other RNA viruses, HCV has a high rate of genetic variability generated by its high mutation rate and the actions of evolutionary forces over time. There are two levels of HCV genetic variability: intra-host variability, characterized by the distribution of HCV mutant genomes present in an infected individual, and inter-host variability, represented by the globally circulating viruses that give rise to different HCV genotypes and subtypes. HCV genetic diversity has important implications for virus persistence, pathogenesis, immune responses, transmission, and the development of successful vaccines and antiviral strategies. Here we will discuss how HCV genetic heterogeneity impacts viral spread and therapeutic control.

## 1. Introduction

The hepatitis C virus (HCV) is an enveloped positive-sense single-stranded RNA virus of the family *Flaviviridae* [1]. Until recently, HCV was the only known member of the genus *Hepacivirus*; however, other *Hepacivirus* RNA genomes have recently been isolated from domestic dogs and horses [2,3]. HCV is predominantly a blood-borne virus [4], with very low risk of heterosexual or vertical transmission. The origin of HCV infection in humans remains unclear, but it may be speculated that arthropod-borne HCV precursors could have been passed to insectivorous small mammals through blood transmission or the ingestion of insects.

HCV was first identified in 1989, when development of the polymerase chain reaction (PCR) enabled construction of a random-primed complementary DNA library from plasma containing what was at that time called a non-A, non-B hepatitis agent [5,6,7]. HCV is hepatotropic, mainly replicating in liver hepatocytes. Following an incubation period of 2–12 weeks, HCV infection starts with an acute phase that usually goes undiagnosed. Acute HCV infection is generally asymptomatic, although it may be associated with jaundice and elevated alanine aminotransferase (ALT) levels. After the acute phase, HCV infection is spontaneously cleared in 18–34% of infected individuals [8].

When HCV infection does not spontaneously resolve, it is known as chronic hepatitis C (CHC). CHC is generally a slowly progressive disease, characterized by persistent hepatic inflammation, and leading to liver fibrosis and cirrhosis development in approximately 10–20% of infected individuals over 20–30 years of HCV infection [9]. Once cirrhosis is established, disease progression is unpredictable. Cirrhosis remains indolent for many years in some individuals, while others exhibit progression to hepatocellular carcinoma (HCC), hepatic decompensation, and death. According to the Global Hepatitis Report, in 2015, 71 million people were chronically infected with HCV, and there were an estimated 1.75 million new infections (global incidence: 23.7 per 100,000) [10]. Additionally, the Global Burden of Disease estimates that 580,000 people died from HCV in 2017 [11].

The HCV particle has a diameter of approximately 68 nm (range: 45–86 nm) [12], and contains a linear positive-sense single-stranded RNA genome encoding 10 viral proteins (Figure 1) [13]. The HCV RNA genome has a length of ~9.6 kb, including a single open reading frame and flanking 5′ and 3′ untranslated regions (UTRs), with a specific structure for viral RNA replication and translation. The 5’-UTR contains the internal ribosome entry site (IRES), from which begins the translation of a polyprotein of ~3000 amino acids. This polyprotein is cleaved by both cellular and viral proteases. Cellular proteases cleave the HCV polyprotein into three structural proteins: core, E1, and E2 [14]. On the other hand, the virus-encoded proteases NS2 and NS3/4A cleave the HCV polyprotein into seven non-structural (NS) proteins: p7, NS2, NS3, NS4A, NS4B, NS5A, and NS5B [14]. The NS5B protein is an RNA-dependent RNA polymerase (RdRP), which synthesizes and replicates viral RNA to produce new viral genomes that are incorporated into the virus particles. HCV NS5B RdRP lacks proofreading activity, resulting in a high error rate during HCV replication, which promotes HCV genetic heterogeneity and population complexity.

The intricacy of the HCV replication cycle in hepatocytes is an important aspect of HCV biology. This complexity may explain why infected sera has not exhibited efficient viral replication in experiments using cultured human hepatoma cells and primary hepatocytes or animal models, apart from chimpanzees. Since its identification in 1989, further investigations of HCV and the development of HCV therapies have been hindered by the absence of a virus replication system. In 1999, the first functional HCV replicon was generated [15]; it was a replicon of genotype 1b, which lacked virus structural proteins, included a selectable marker, and replicated in human hepatoma cells (Huh7). In 2005, the first infectious virus particle was obtained—a genotype 2a isolate (JFH-1) that could grow in cell culture [16]—finally enabling recapitulation of the entire HCV life cycle in cell culture. Since then, the enormous amount of data generated regarding the HCV life cycle has greatly improved our knowledge of virus biology in general, and particularly our understanding of the genetic heterogeneity and complexity of HCV.

## 2. HCV Quasispecies and Diversity

The term viral quasispecies refers to a population structure that includes extremely large numbers of variant genomes, i.e., a mutant spectra [17,18]. The quasispecies concept was originally developed to explain the self-organization and adaptability of primitive replicons as an essential step in the origin of life [19], and has recently been adopted to interpret the adaptive potential of RNA viruses. Clonal analyses and virus genome sequencing have demonstrated that RNA virus populations comprise a myriad of different mutant genomes (Figure 2). This mutant complexity could explain their adaptive behavior, which impacts virus transmission, tropism, spread, pathogenesis, and host control.

Unlike DNA polymerases, RNA virus RdRPs lack proofreading activity, yielding an estimated error rate of 10^−3^ to 10^−5^ mutations per nucleotide per replication cycle [17,18]. During acute infection an organism may contain 10^9^–10^12^ viral particles at any given time. CHC patients also exhibit high serum viral loads of 10^5^–10^7^ virions/mL. With an RNA virus genome length of approximately 10^4^ nucleotides, it is likely that every possible single mutant and many double mutants will occur by the time the population reaches the size of many natural virus populations (Figure 2). Notably, RNA virus mutation rates are closely related to genome length. For instance, the *Coronaviridae* family of RNA viruses has the largest genomes (26–32 kb), and these viruses proofread and remove mismatched nucleotides during genome replication and transcription through a virus codify exonuclease proofreading function [20]. This relationship indicates that the mutation rate of an RNA virus is an adaptive trait that is near an unsurpassable threshold.

As in many RNA viruses, genetic variability is a prominent feature of HCV [21]. Experimental evidence demonstrates that HCV populations comprise a distribution of mutant genomes, i.e., a quasispecies [22,23,24]. This high genetic diversity of HCV is supported by its high replication kinetics (10^12^ virions per day), and the low fidelity of its RdRP, similar to other RNA viruses. These factors contribute to a high rate of viral mutation, with 10^−4^ to 10^−5^ mis-incorporations per copied nucleotide and replicative cycle, such that HCV genomes accumulate an estimated 0.3–1.2 nucleotide substitutions per cell infection [25,26]. This high mutation rate, together with evolutionary forces acting over time (selective pressure, recombination, and genetic drift), have yielded two levels of HCV genetic heterogeneity: intra-host variability, i.e., the virus variant cloud (quasispecies) present in an infected individual; and inter-host variability, i.e., the worldwide circulating viruses that give rise to different virus genotypes and subtypes.

Sub-Saharan Africa exhibits extensive genetic inter-host heterogeneity of HCV, strongly suggesting that this virus was endemic to this geographical area long before its global spread over the last 100–200 years. Based on genomic sequence variations, HCV has been classified into 8 genotypes and 105 subtypes [27,28]. Although the global spread of HCV likely preceded the human immunodeficiency virus type 1 (HIV-1) epidemic by several decades, both were relatively recent events. Nevertheless, HCV global diversity is greater than that of HIV-1. It has been hypothesized that HCV diversity is influenced by other parameters of the HCV life cycle, such as the long-lived nature of HCV-infected cells compared to HIV-1-infected cells, the existence of multiple replication complexes within an infected cell that may permit the accumulation of diversity, and the turnover rate of both the replication complexes and infected cells [26].

The different HCV genotypes, named from 1 to 8 in order of their discovery, differ from each other by 30–35% of their nucleotide sequence. These genotypes are further divided into subtypes, defined by letters (1a, 1b, 2a, 2b, 3a, etc.), which differ from each other by 20–25% of their nucleotide sequence. Isolates within the same subtype can differ by 10%. Although the different genotypes and subtypes share basic biological and pathogenic characteristics, they differ in their epidemiology and responses to treatment (see below).

The prevalence rates and distributions of the different genotypes and subtypes vary according to geographical area [29]. Genotype 1 is the most prevalent worldwide, with subtype 1a showing a higher prevalence in the United States and Canada, and subtype 1b more prevalent in Europe. Genotype 2 is predominant in West Africa, while genotype 3 is endemic to Southeast Asia. Genotype 4 is mainly found in the Middle East, Egypt, and Central Africa. Genotype 5 occurs almost exclusively in South Africa, while genotype 6 is predominantly distributed throughout Asia. The recently discovered genotype 7 was identified in seven infected individuals in the Democratic Republic of the Congo, and genotype 8 was found in four infected individuals from Punjab (India). Worldwide, the most prevalent genotypes are 1 and 3, respectively comprising 44% and 25% of HCV infections.

HCV genotype distribution also varies according to the epidemiology of infected individuals, e.g., intravenous drug users (IDUs), hemophiliacs, and men who have sex with men. For instance, genotype 4 is predominant in the Middle East, Egypt, and Central Africa, but now also shows a high prevalence among IDUs in southern Europe. As will be discussed later in this review, HCV genotype and subtype diversity affect the clinical efficacy of antiviral medications. Naturally occurring intra or intergenotypic recombinants are rare; however, a specific 1b/2k intergenotypic recombinant has spread widely enough to become epidemiologically important [30].

HCV intra-host quasispecies genetic diversity is estimated to be over 1–3%, but is not equally distributed throughout the HCV genome. The highest diversity is observed in the structural proteins, which are most exposed to immune surveillance and pressure—for example, the hypervariable region 1 (HVR1) at the N-terminus of E2 [31]. In contrast, NS virus proteins are less heterogeneous, largely because they must maintain their enzymatic activities [22]. The application of ultradeep sequencing to a modified replicon system enabled the scoring of >15,000 spontaneous mutations, encompassing over 90% of the HCV genome. This revealed >1000-fold differences in mutability across genome sites, with extreme variations even between adjacent nucleotides [32].

Some infected patients have exhibited an exceptional level of heterogeneity in the molecular evolution of HCV over time [33]. This quasispecies heterogeneity includes significant fluctuations in viral genetic diversity over the course of infection, as well as unusual phylogenetic topologies containing multiple distinct lineages that coexist for long time periods. Diversity patterns are associated with the long-term maintenance of viral lineages within patients, which fluctuate in terms of their relative frequency in peripheral blood. These findings demonstrate that HCV replication behavior is complex, and likely involves multiple viral subpopulations with distinct evolutionary dynamics, which may originate rapid fluctuations in viral diversity and the reappearance of viral strains years after their initial detection [33]. The high within-host evolutionary heterogeneity of HCV has important implications for molecular epidemiological analyses. The intermittent detection of diverse lineages in serum means that the obtained HCV consensus sequence may be highly dependent on when sampling occurs, and may not be representative of the virus that is transmitted (Figure 2). This may explain the difficulties faced when trying to reconstruct the evolutionary history of HCV [33,34].

## 3. Implications of HCV Diversity in Pathogenesis and Transmission

Studies of serially sampled HCV sequences have also indicated a link between viral evolution and disease progression. CHC may follow a mild and stable disease course, or may progress rapidly to cirrhosis and liver-related disease and death. Analyses of serial prospectively collected samples from cases of transfusion-associated hepatitis C have revealed that rapidly progressive disease is correlated with greater viral quasispecies diversity and divergence, and a higher rate of synonymous substitution [35]. Throughout the first 7 years of infection, rapidly progressive cases showed higher average substitution rates across all studied virus envelope regions (E1, HVR1, and E2) compared with slowly progressive cases [35]. Rapidly progressive cases exhibited a greater mean overall number of substitutions per site, and this effect was especially pronounced for synonymous substitutions. These findings suggest that faster disease progression is associated with shorter viral generation times, as has also been reported for HIV-1 [36].

Similarly, the evolutionary dynamics of the HCV quasispecies during the acute phase of hepatitis C might predict whether the infection will resolve or become chronic [31]. In an analysis of HCV E1 and E2 envelope gene sequences during the acute phase of HCV infection in 12 patients with different clinical outcomes, acute resolving hepatitis was associated with relative evolutionary stasis of the viral quasispecies population, whereas progressing hepatitis was correlated with HCV genetic evolution [31]. Phylogenetic analysis of all the HVR1 amino acid sequences from each patient at different time-points revealed two topological patterns according to the disease outcome. Resolving hepatitis cases exhibited a generally monophyletic population, with intermingling of sequences derived from different time-points. In contrast, progressing hepatitis cases showed a tendency of cluster formation over time, with branch lengths consistently longer than those observed in acute resolving hepatitis [31]. Additional studies have shown that viral persistence results from a rapidly evolving viral quasispecies, a humoral response that drives viral mutation but is insufficient to neutralize the replicating viral strain, and a T-cell response that is exhausted by combating a high antigenic burden that was fully entrenched prior to effective T-cell mobilization [37]. Remarkably, this early neutralizing containment predicted a slow clinical evolution of CHC [35].

After cirrhosis development, CHC patients carry a 1–5% annual risk of HCC and a 3–6% annual risk of hepatic decompensation, which can lead to orthotopic liver transplantation or liver-related death. Quasispecies analysis of samples from patients with HCV-associated HCC, who underwent orthotopic liver transplantation or partial hepatectomy, revealed that HCC-containing livers harbored a more complex viral population with significantly higher genetic diversity compared to cirrhotic livers without HCC [38,39]. Notably, E1, HVR1, and E2 quasispecies diversity was significantly higher in the livers of patients with HCC compared to controls with non-HCC cirrhosis. Genetic diversity, both within and outside of the HVR1, was also higher in the serum of patients with HCC. While the overall viral diversity did not differ between the tumor and surrounding non-tumorous tissues, characterization of individual virus variants revealed changes in the viral population between tumors and non-tumorous areas, suggesting HCV compartmentalization within the tumor [39,40]. This compartmentalization was not observed between non-tumorous areas and serum, or between different areas of the cirrhotic liver or between liver and serum in control patients. However, liver virus compartmentalization remains somewhat controversial. In a study of end-stage liver disease patients undergoing liver transplant, no intra-hepatic E1/E2 quasispecies compartmentalization was observed [41].

The mechanism by which HCV gains entry into cells is complex, involving a broad range of host proteins. Entry is a critical phase of the viral life cycle, and may impact virus diversity and evolution over time. Molecular identification of transmitted/founder HCV genomes that lead to productive clinical infection is essential for elucidating key aspects of HCV transmission, biology, immune-pathogenesis, and natural history. Moreover, understanding the acute infection period is critical for vaccine development since the vaccine must target transmitted viruses.

Deep sequence analysis of half of the HCV genome (including genes encoding E1, E2, P7, NS2, and NS3), from the initial HCV RNA-positive sample from patients with acute infection (6–8 weeks), revealed that the number of transmitted viruses leading to productive clinical infection ranged from 1 to 37 or more (median 4) [42]. During the first weeks following virus transmission, HCV sequence evolution generally conforms to a simple model of random virus evolution, with sequences characterized by a within-lineage star-like phylogeny and a Poisson distribution of mutations [42,43]. Following initial virus transmission, the average sequence diversity increases almost linearly at early time-points during the virus titer exponential growth phase, and then later becomes saturated when the virus reaches the plateau phase [26,42]. After the emergency release of neutralizing antibodies and cytotoxic T cells, the viral load decreases and non-random virus diversity becomes evident. Another study analyzed a larger dataset of deep-sequenced full-length viruses, obtained from subjects with acute HCV infections, and identified a single transmitted founder in 26% of samples [44]. The presence of a single transmitted founder virus was not associated with any host or virus-related factors, notably including viral genotype and spontaneous clearance [44,45]. Although it cannot be excluded that viral factors may influence spontaneous HCV clearance, several host factors seem to be implicated in this process. Time since infection, host gender, and continent of origin have not been associated with genomic heterogeneity during HCV acute infection [45].

## 4. HCV Therapy

Quasispecies diversity presents challenges to host immune surveillance (including neutralizing antibodies and activated T-cells), antiviral therapies, and effective vaccine development. Despite immune activation, HCV can often evade the host response and establish persistent infection [46]. There is a consensus regarding the defective and not maintained adaptive immunity present in patients that progress to CHC [47]. The main challenge to developing an effective preventive HCV vaccine is that HCV does not elicit protection against reinfection. Experimental in vivo studies have demonstrated that chimpanzees with resolved acute HCV infection could develop hepatitis again when re-challenged with the homologous strain [48]. However, in most cases, re-challenged animals have exhibited protection against developing CHC, indicating some degree of protective immunity [49]. Similarly, CHC patients who exhibit viremia resolution following antiviral therapy do not acquire protective immunity, and can thus experience reinfection [47]. Reinfections can even occur with the same HCV subtype [50], suggesting the difficulty of designing a prophylactic HCV vaccine. It remains unclear whether the quasispecies nature of HCV populations is related to the lack of protective immunity after CHC; however, the development of a prophylactic vaccine must account for the huge genetic heterogeneity of HCV.

The absence of data about the immunological determinants of viral clearance, persistence and protective immunity is a major barrier to the development of a prophylactic HCV vaccine [47]. Recent work has shown that HCV variants isolated pre-seroconversion are more sensitive to the antiviral activity of first line innate immune response represented by interferon-induced transmembrane proteins than variants from patients isolated during CHC post-seroconversion [51]. These results indicate that interferon-induced transmembrane proteins are drivers of viral immune escape and antibody-mediated HCV neutralization in acute HCV infection. Similarly, after primary infection, an incomplete control of HCV replication in the absence of adequate CD4+ T cell help is associated with the emergence of viral escape mutations within the major histocompatibility complex (MHC) class I epitopes targeted by virus-specific CD8+ T cells [52]. In addition to CD8+ T cell exhaustion, the main effector cells in HCV infection, failure of T cell immunity has been also attributed to HCV escape. Viral escape mutations usually occur during the first 6 months of infection [53] and they may affect about half of the CD8+ T cell-targeted epitopes [54]. Moreover, the occurrence of viral escape mutations during CHC is uncommon and may reflect the absence of T cell-mediated selection pressure during this period. An important aspect of HCV escape mutations to specific CD8+ T cells is the impact that they have en the virus replication capacity. This reduction in the virus fitness leads to the appearance of compensatory substitutions or to their rapid reversion when the selective immune pressure disappears. The fitness cost of these escape mutations may explain why some specific HLA-types (e.g., B27, A57 or A3) are associate with a high probability of HCV clearance after acute infection [55,56]. CD8+ T cell responses to linked with these HLAs usually target conserved virus genome regions and escape mutations are not easily tolerated. These conserved epitopes are clear targets for the development of an effective HCV T cell-based vaccine [57]. Correlates of protective immunity to overcome HCV genetic variability have to be described in order to design an effective HCV vaccine.

In contrast to the difficulties in finding a HCV preventive vaccine, HCV antiviral development has been successful. Since the development of direct-acting antiviral (DAA) therapy (Table 1) for CHC in 2014, cure rates have increased to around 95%. DAA therapy is among the best examples of success in the fight against viral infections. DAAs have transformed HCV management, and can accelerate the global eradication of HCV.

A CHC patient is considered cured of HCV infection once a sustained virological response (SVR) is achieved. SVR is defined as undetectable HCV RNA in serum or plasma at 12 or 24 weeks after completing treatment, based on a sensitive assay with a lower limit of detection of ≤15 IU/mL [58]. In 1997, interferon alfacon-1 (Infergen^®^) was the first drug approved by the first US Food and Drug Administration (FDA) for use against HCV infections (Figure 2). However, this drug was discontinued in 2013 due to severe adverse events. Over time, the arsenal to combat HCV infections has grown to include ribavirin (Copegus^®^, Rebetol^®^, Virazole^®^) in 1998, pegylated interferon 2b alfa (PegIntron^®^, Intron^®^-A) in 2001, and pegylated interferon 2a alfa (Pegasys^®^, Roferon^®^-A) in 2002 [59].

Basic research on HCV has untangled critical components of the complex virus life cycle, which has greatly contributed to the development of highly effective DAA (Table 1). DAAs provide a cure for HCV in most individuals [60]. Between January 2011 and November 2016, ten therapies were approved by the US FDA, and another two were approved in Japan (asunaprevir plus daclatasvir, and vaniprevir plus ribavirin plus pegylated interferon alfa) [59]. In 2011, the first-generation NS3/4A protease inhibitors telaprevir (TVR) and boceprevir (BOC) became the first DAAs approved for use in combination with pegylated interferon and ribavirin, for the treatment of HCV genotype 1. The SVR rates in genotype 1 infections were between 65–75% [61,62]; however, both TVR and BOC were discontinued due to severe side effects and commercial considerations. In 2013, the NS3/4A protease inhibitor Simeprevir (SMV) was approved for use in combination with pegylated interferon and ribavirin for the treatment of genotype 1 infection. Compared to its predecessors, SMV achieved comparable SVR rates but with improved tolerability [63].

A landmark in HCV infection treatment was the development of the NS5B polymerase inhibitor sofosbuvir (SOF). SOF is a nucleotide analogue that produces early chain termination following its incorporation into newly synthesized viral RNA [59]. SOF targets the conserved active site of NS5B, and is thus active against all HCV genotypes and has a high barrier to resistance. Although SOF resistance can be produced in tissue culture, it has rarely been observed in vivo. Moreover, viruses containing the SOF resistance mutation (NS5B at position S282T) exhibit very low replication capacity. SOF combined with pegylated interferon 2a alfa and ribavirin achieved an SVR rate of 90% after 12 weeks of therapy in individuals infected with genotypes 1 and 4 [64]. Similarly, an oral regimen of SOF plus ribavirin achieved SVR rates of 95% and 82% after 12 weeks of therapy in naive and treatment-experienced patients infected with genotypes 2 and 3, respectively [65]. Thus, in 2013, the FDA approved SOF for use in combined therapy with pegylated interferon 2a alfa for HCV genotypes 1 and 4, and with ribavirin alone for genotypes 2 and 3.

The successful combination of SOF with ribavirin ushered in the era of interferon-free regimens. In 2014, the combination of SOF with the NS5A inhibitor ledipasvir (LDV) was approved as a once-daily co-formulation for treating HCV genotype 1. This combination aimed for rapid suppression of viral replication, and prevention of resistant variant selection. The SOF/LDV combination was evaluated with and without ribavirin, achieving SVR rates of 94–99% after 12 weeks of therapy [66,67,68]. In 2014, the combination of SOF plus SMV was approved for treatment of HCV genotype 1 [69]. Moreover, at the end of 2014, approval was granted for combinations of ombitasvir, ritonavir-boosted paritaprevir, and dasabuvir ribavirin, which achieved SVR rates of >90% after 12 or 24 weeks of treatment in HCV genotype 1-infected individuals with or without cirrhosis [70,71]. Paritaprevir is an NS3/4A protease inhibitor, ombitasvir is an NS5A inhibitor, dasabuvir is a non-nucleoside NS5B polymerase inhibitor, and ritonavir is an HIV-1 protease inhibitor that can boost other protease inhibitors of different families. More recently, approval has been granted to different combinations of DAAs that exhibit increased potency and pan-genotypic efficacy—including SOF plus velpatasvir (VEL; NS5A inhibitor), elbasvir (NS5A inhibitor) plus grazoprevir (NS3/4A inhibitor), glecaprevir (NS3/4A inhibitor) plus pibrentasvir (NS5A inhibitor), and SOF/VEL plus voxilaprevir (NS3/4A inhibitor).

Real-world clinical experience has confirmed the results of various DAA clinical trials. Studies have included patients infected with all HCV genotypes, and with advanced liver disease. Importantly, regimens including only 8 weeks of treatment have achieved SVR rates of over 95% [72,73]. Since the new DAA therapies can achieve SVR in >95% of subjects irrespective of HCV genotype, genotype testing is no longer required before starting treatment. This supports the wider prescription of DAAs, especially in resource-limited environments where genotype testing can be challenging and cost-prohibitive. Another important feature is that new DAAs are highly effective among special populations, including individuals with chronic kidney disease or hemoglobinopathies, HIV-1/HCV-co-infected individuals, IDUs, older individuals, and patients with advanced liver disease. Recent studies have described new NS5B nucleotide analogues, including guanosine [74] and uridine analogues [75], and non-nucleoside inhibitors [76,77]. New pan-genotypic NS5A inhibitors have been also described [78,79,80]. The aim in developing new DAAs is to obtain compounds with a broader capacity to inhibit the different genotypes and HCV variants, and having a better metabolic profile [81].

## 5. DAA Resistance

Although current antiviral combinations can cure a large percentage of HCV-infected patients, the presence of resistance-associated substitutions (RASs) may reduce the success of antiviral therapies. RASs are particularly a concern during the re-treatment of patients treated with DAAs. However, the complexity of HCV quasispecies allows this virus to rapidly explore the available sequence space and, thus, it is not surprising that (RASs) can exist in the mutant spectra even in DAA-naïve HCV-infected patients [82].

The impact of HCV diversity on RAS development has been clearly demonstrated by the European HCV Resistance Study Group, which performed an exhaustive analyses of RAS patterns in a cohort of 626 non-responder/breakthrough and relapsed patients, compared to 2322 DAA-naïve patients infected with HCV genotypes 1 to 4 [83]. They identified resistance patterns that differed depending on the HCV genotype and subtype, and the target regions of the utilized drug combinations (Table 2) [83,84,85]. Infrequent subtypes in Europe, North America, Japan and Australia (e.g., 1l, 4r,3b, 3g, 6u or 6v) harbor natural polymorphisms that confer inherent resistance to NS5A inhibitors [86]. Another example of how HCV subtype may affect therapy outcome is shown by the suboptimal effectivity of SOF/VEL in patients with compensated cirrhosis infected with HCV genotype 3a carrying the Y93H RAS in theNS5A region of the viral genome [87].

Previous experience with HIV-1 antiretroviral resistance has been useful for overcoming the problem of HCV drug resistance with the new DAA-based therapies. Similar to anti-HIV-1 cocktail therapies, combination therapies have been developed that target different stages of the HCV life cycle with the aim of avoiding DAA resistance. In addition to quasispecies complexity, another factor that contributes to resistant variant development is the genetic barrier to drug resistance, which is based on the number and types of mutations needed for RAS emergence. The fitness of the resistant variant population is also critical, as it determines the likelihood that a resistant variant will persist within the larger viral quasispecies population.

In contrast to the nucleotide/nucleoside analogues used for HIV-1 therapy, the leading nucleotide analogue employed in HCV DAA therapies, SOF, presents a higher barrier to drug resistance. Moreover, SOF RASs (e.g., SOF RAS S282T) have a negative impact on HCV fitness. These factors have greatly minimized the general problem of resistance to the new DAA therapies. The emergence of viral resistance is also influenced by the level of drug exposure, with suboptimal concentrations of antiviral agents potentially leading to RAS selection by allowing maintenance of a viral load in the presence of mild selective pressure. Since HCV can be eradicated from infected individuals, HCV therapies are administered for a limited time period, in contrast to the life-long administration of HIV-1 antiretroviral therapies. Therefore, the drug toxicity of HCV therapies may be less relevant, as it is possible to use a higher drug dosage for a limited period of time (e.g., 8 weeks).

From the start of DAA therapy implementation, there has been growing interest in identifying pre-existing RASs that are present within HCV populations. Several methods have been applied to perform sequencing to varying depths, with the aim of identifying population bulk viral variants, and minority low-frequency variants present in HCV quasispecies [89,90]. Most of these studies have performed population sequencing via the traditional bulk Sanger method; however, this strategy lacks sensitivity and generally cannot detect viral populations constituting less than 10–25% of the total population [82] (Figure 2). The development of high-throughput next-generation sequencing technologies has rapidly improved our ability to detect viral subpopulations constituting ever-smaller proportions of HCV populations, even identifying variants with frequencies of 0.1–1%. In general, RASs that are present in low proportions (<15%) do not significantly affect treatment outcomes, whereas RASs existing in greater than 15% of the overall population are more commonly associated with treatment failure. Thus, it is agreed that a 15% cut-off should be used for reporting RASs by population and next-generation sequencing in all clinical trials, and in studies of real-world infected individuals [72].

The pan-genotype potency of new DAA therapies is expected to reduce the incidence of treatment failure due to baseline RASs. However, re-treatment of patients who previously failed a DAA-based therapy can be optimized based on RAS testing. It should be noted that the two newer pan-genotype regimens approved for treating HCV genotypes 1–6 (glecaprevir/pibrentasvir and SOF/VEL/voxilaprevir) each include an NS3/4A protease inhibitor (glecaprevir or voxilaprevir). It was recently demonstrated that NS3 position 156 is a hotspot for RASs among genotypes 1–4 (but not for genotypes 5 and 6), and confers resistance to glecaprevir or voxilaprevir [91]. Individuals who fail to respond to DAA therapies have exhibited RASs at NS3-156, which might reduce the clinical effectiveness of these new combination treatments. In this contest, clinical guidelines recommend HCV resistance testing prior to retreatment in patients who failed after any of the DAA-containing treatment regimens [88]. The resistance profile observed in these resistance tests can guide retreatment. These tests, in addition to HCV subtype genotyping may allow the treatment with less expensive, non-pan-genotypic DAA combinations. Nevertheless, two phase III trials have demonstrated the safety and efficacy of the triple combination of SOF/VEL/voxilaprevir in patients who previously failed to achieve SVR with a DAA-based regimen, including patients exposed to protease and/or NS5A inhibitors [92]. In first of these trials, the overall retreatment SVR rate was 96%. Neither the HCV genotype, nor the presence of resistance mutations at retreatment baseline had an influence on the response. Within the patients with virological failure the following RASs were identified at baseline: NS3 Q80K and NS5A A30K. L30R, Q30T and Y93NY/H. The second trial also included patients who had previously failed to achieve SVR following a DAA-based treatment, but this time, the failing treatment did not include an NS5A inhibitor [92]. The overall retreatment rate was 98%. Again, neither the HCV genotype, nor the resistance profile at retreatment baseline had an influence on the response in patients receiving the triple combination. Several real-world studies have confirmed the efficacy of SOF/VEL/voxilaprevir in the retreatment of previous failures of treatments that contained DAAs. In contrast to the results obtained with the former triple combination, the combination of glecaprevir/pibrentasvir showed a lower rate of SVR in patients previously exposed to a NS5A inhibitor [93]. Treatment failed in 7.3% of patients with genotype 1a infection. Failing patients selected RASs in the NS3 and NS5A regions. In brief, the combination of glecaprevir and pibrentasvir should be avoided in the retreatment of patients who failed a prior DAA-containing regimen, particularly if this regimen contained an NS5A inhibitor. As an alternative, a triple combination of SOF with an NS3 protease inhibitor and an NS5A inhibitor appears to be better suited to retreatment of DAA exposed patients [88]. Since pibrentasvir has an in vitro higher barrier to resistance than all other approved NS5A inhibitors [94], a triple combination with this DAA may be an interesting alternative for retreatment of difficult-to-cure patients [88].

## 6. Conclusions and Perspectives

The increased availability of DAA therapy for HCV infection is starting to impact the morbidity and mortality rates. Western countries have seen decreased numbers of HCV-infected individuals on the liver transplantation waiting list [95]. DAA treatment is also correlated with reductions in HCC, liver-related mortality, and overall mortality in patients with cirrhosis [96]. Curative treatment with DAAs is important because cured people do not transmit the infection. Eliminating CHC will require the provision of DAA treatment to all people living with HCV. There remains a need for comprehensive strategies for testing and diagnosis, treatment initiation, and care follow-up. Epidemiological data collection should be expanded to more accurately characterize high-risk populations. The development of new second generation of DAAs with safe a well-tolerated profiles is allowing the prescription of DAA combinations that are yielding high rates of HCV cure (i.e., 96–98%). However, the complex HCV quasispecies dynamics expected from low-fidelity replication can allow the selection of DAA RASs that may significantly impact treatment outcomes [72,97]. In very difficult-to-cure patients, who failed twice o more to achieve SVR after a combination regimen that included a protease and/or an NS5A inhibitor, the triple potent pan-genotypic combinations of SOF/VEL/voxilaprevir or SOF/glecaprevir/pibrentasvir are showing promising results, even with patients that have DAA RASs at baseline of retreatment [88].

## Figures and Tables

**Figure 1 viruses-13-00041-f001:**
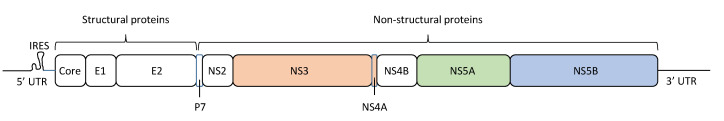
Hepatitis C virus (HCV) genome structure. The HCV RNA genome encodes a protein of ~3000 amino acids in length, which is cleaved to generate three structural proteins (core, E1, and E2) and seven non-structural (NS) proteins (p7, NS2, NS3, NS4A, NS4B, NS5A, and NS5B). In this figure, the amino acid positions of these proteins are mapped, and the 5’ untranslated region (5’-UTR) and 3′ untranslated region (3’-UTR) are indicated. Approved antiviral agents directly target NS3/4A, NS5A, and NS5B for effective inhibition of HCV replication.

**Figure 2 viruses-13-00041-f002:**
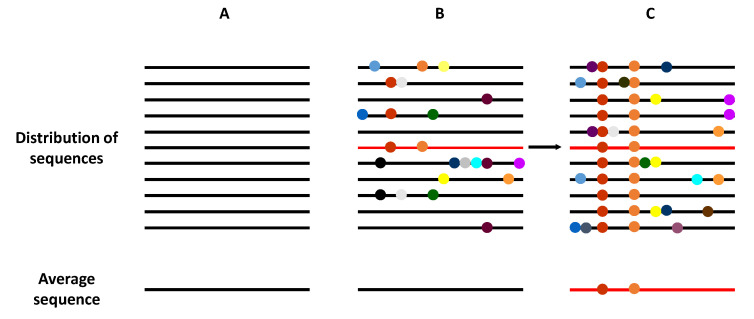
Schematic representation of the genome composition of a virus population. Viral genomes are represented as horizontal lines, and mutations as symbols in the lines. (**A**) A homogeneous virus population in which no mutations are generated, such that all genomes within this population are identical. (**B**) A heterogeneous virus population in which mutants are generated at a high frequency, such that most of the genomes are genetically different. However, the average consensus sequence of this population is identical to in the homogeneous population (**A**). (**C**) The selection of an individual genome (marked in red), via random drift or positive selection, generates a new distribution of variants if the replication system produces new mutations. Now, the average population genome sequence is different from that shown in panels (**A**,**B**).

**Table 1 viruses-13-00041-t001:** Summary of approved HCV direct acting antivirals (DAAs).

DAA Virus Target	DAA Name
**NS3/4A inhibitor**	Telaprevir ^a^
	Boceprevir ^a^
	Semiprevir
	Paritaprevir
	Grazoprevir
	Voxilaprevir
	Glecaprevir
**NS3/4A booster**	Ritonavir
**NS5A inhibitor**	Ledivaspir
	Ombitasvir
	Daclatasvir
	Elbasvir
	Velpatasvir
	Pibrentasvir
**NS5B inhibitor**	Sofosbuvir
	Dasabuvir

^a^ Discontinued.

**Table 2 viruses-13-00041-t002:** Substitutions associated with resistance to HCV direct acting antivirals (DAAs).

Genome Region Drug Class ^a^	Amino Acid Position	Genotype						
		1a	1b	2	3	4	5	6
**NS3 Protease Inhibitors:** Semiprevir, Paritaprevir, Grazoprevir, Voxilaprevir and Glecaprevir ^b^								
	36	V36A/C/F/G/L/M	V36A/C/G/L/M					V36I
	41	Q41R	Q41R		Q41K	Q41R		Q41K/R
	43	F43I/L/S/V	F43I/S/V	F43V				
	54	T54A/S	T54A/C/G/S					
	55	V55I	V55A	V55A/I				
	56	Y56H	Y56H/L/F	Y56H/F	Y56H	Y56H		Y56H
	80	Q80K/L/R	Q80H/K/L/R		Q80K/R	Q80R		L80K/Q
	122	S122G/N/R	S122A/D/G/I/N/R/T					S122T
	155	R155G/I/K/M/Q/S/T/V/W	R155C/G/I/K/L/Q/M/S/T/W		R155K	R155C/K	R155K	
	156	A156G/P/S/T/V	A156G/P/S/T/V	A156L/M/T/V	A156G/P/T/V	A156G/H/K/L/S/T/V	A156T/V	A156T/V
	158	V158I	V158I					
	166				A166S/T/Y			
	168	D168A/C/E/F/G/H/I/K/L/N/Q/R/T/V/Y	D168A/C/E/F/G/H/I/K/L/N/Q/T/V/Y	D168A/E/F/G/H/N/S/T/V/Y	Q168H/K/L/R	D168A/E/G/H/T/V	D168A/E/H/K/R/V/Y	D168A/E/G/H/V/Y
	170	I/V170T/V	I/V170A/L/T					I170V
	175		M175L					
**NS5A Inhibitors:** Ledivaspir, Ombitasvir, Daclatasvir, Elbasvir, Velpatasvir and Pibrentasvir								
	24	K24E/QR/T	Q24K	T24A/S	S24F			
	26	K26E						
	28	M28A/G/S/T/V	L28A/M/T	L/F28C/S	M28T/K	L28M/S/T/V	L28I	F/L28A/I/L/M/T/V
	29	29 P29R	P29S, del29	P29S				
	30	Q30C/D/E/G/H/K/L/N/R/T/Y, del30	R30G/H/P/Q/S	L30H/S	A30D/E/K/S	L30F/G/H/R/S	Q30H	R30E/H/N/S
	31	L31I/F/M/P/V	L31F/I/M/V/W	L31I/M/V	L31F/I/M/P/V	M/L31I/V	L31F/I/V	L31I/M/V
	32	P32L/S, del32	P32F/L/S, del32				P32L	P32A/L/Q/R/S
	38	S38F						
	58	H58C/D/L/P/R	P58A/D/L/S/R/T			T58A/P/S		T58A/G/H/N/S
	62		Q/E62D		S62L			
	92	A92K/T	A92E/K/T/V	C92R/S/T/W	E92K			E92T
	93	Y93C/F/H/L/N/R/S/T/W	Y93C/H/N/R/S/T	Y93F/N/H	Y93H/N/S	Y93C/H/N/S/R/W		T93A/H/N/S
**NS5B Non-nucleoside Inhibitors:** Dasabuvir								
	314	L314H						
	316	C316Y	C316H/N/Y/W					
	368		S368T					
	395	A395G						
	411							
	414	M414I/T/V	M414I/T/V					
	445		C445F/Y					
	446	E446K/Q						
	448	Y448C/H	Y448C/H					
	553	A553T/V	A553V					
	554	G554S	G554S					
	555	Y555H						
	556	S556G/R	S556G/R					
	557	G557R						
	558	G558R	G558R					
	559	D559G/N	D559G/N					
	561	Y561H/N						
	565	S565F						
**NS5B Nucleotide analogue Inhibitors:** Sofosbuvir								
	150				A150V			
	159	L159F	L159F	L159F	L159F			
	206				K206E			
	282	S282G/R/T	S282G/R/T	S282G/R/T	S282G/R/T	S282C/G/R/T	S282G/R/T	S282G/R/T
	316	C316H/R	C316F/H/N					
	320	L320I/F/V						
	321	V321A	V321I		V321A	V321A		

^a^ The resistance-associated substitutions (RASs) showed here were adapted and updated from [84,88]. ^b^ First-generation protease inhibitors telaprevir and boceprevir have been excluded of this analysis.

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
