# Peer review of "Therapy Implications of Hepatitis C Virus Genetic Diversity"

_viruses, 2020, doi:10.3390/v13010041_

Round 1
Reviewer 1 Report
This review manuscript by Miguel Angel Martinez and Sandra Franco summarizes the impact of the antiviral therapy on the genetic diversity of HCV during antiviral therapy and elaborates on how HCV genetic heterogeneity impacts viral spread and therapeutic control. The author introduce nicely the genetic diversity and quasispecies of HCV, its pathological consequences and summarize the latest regiments of direct-acting antivirals together with the current knowledge of escape mutations. The authors provided a comprehensive, up-to-date and well written manuscript with only minor adjustments required (see minor comments below). Despite efficient antiviral treatment is available the topic of the review remains relevant with high interest to the field.
Minor comments:
Line 33: Indeed, Michel Houghton first cloned the HCV genome but in light of the 2020 Nobel prize for the discovery of HCV, the other two laureates Harvey Alter and Charles Rice should be cited as well and the sentence rephrased accordingly.
Line 220: Additionally Fafi-Kremer et al (PMID 20713596) should be cited demonstrating a selection of HCV quasispecies after liver transplantation with impact on reinfection and viral evasion from neutralization.
Line 273: This statement (“DAAs have transformed HCV management, and have paved the way to global eradication of HCV”) is too optimistic and should be downtoned a bit: So far, no virus has been eliminated only by antiviral treatment without an efficient vaccination. Moreover, HCV infection is often asymptomatic and diagnosed late. Together, with the still high costs of DAA cure this argues against a global eradication of HCV especially in countries with limited resources and weak heath care systems.
Figure 1: typo “Acute infection”
Author Response
Please see attached file with our point-by-point response to your comments.

Reviewer 2 Report
The submitted manuscript is a review about the therapeutic implications of intra-host and inter-host viral genome heterogeneity. Although the topic may be interesting, the authors focus their attention on aspects not related to therapy (eg: natural history, geographical distribution) or not current (eg: IFN-based therapeutic regimens). Moreover, the history of antiviral therapy appears to be decontextualized, outdated and far from the objectives of the work. This makes the overall review unattractive and reduces its interest. Moreover, the title appears to describe a marginal part of the work and does not appear to represent it fully. Finally, the most interesting part of the work in my opinion (resistance to DAAs) has been addressed only marginally and should be implemented. For all these reasons, I believe that this work needs major revisions in order to be reconsidered for publication.
Major revisions:
- Title is not representative of the paper in the way it was written. Since the therapeutic implications of genetic diversity of the virus are of indisputable interest, it is advisable to reformulate the work (rather than the title) by focusing attention on the therapeutic aspects. In fact, as it is structured, this review appears not very current in an era in which the new highly effective therapeutic regimens overcome many of the issues raised in the work. In particular, the issue of resistance to DAAs is in my opinion implementable and would make the work of greater interest. It might be useful to focus more attention on the role of resistances in therapeutic failures to second generation DAAs and on the effectiveness of some therapeutic regimens (eg: voxilaprevir) in overcoming such failures. In fact, only a few lines of the paper are dedicated to the latter.
- Genomic heterogeneity is the cause of the difficulties in preparing a vaccine. This topic is therefore absolutely congenial to the paper, but in my opinion it has been treated only marginally. It is therefore advisable to dedicate a specific paragraph to the vaccine and implement its contents.
- For the aforementioned reasons, the conclusions should be modified and made faithful to the topic (therapeutic implications).
Minor revisions:
- The discussion on IFN-based regimes appears out of date. Therefore a drastic downsizing is recommended.
- Similarly, in my opinion, the long discussion on the natural history of the disease goes beyond the topic. Therefore, a drastic resizing and elimination of figure 1 is recommended.
Author Response

(The authors gave the same response as above.)

Round 2
Reviewer 2 Report
I thank the Authors for accepting the advice. In my opinion, the paper in the last draft has acquired greater interest and it’s more current. There is also a good correspondence between the title of the work and the text, unlike the first draft. The revision requests appear completely resolved. For these reasons, in my opinion the manuscript can be considered for publication on "Viruses".